# Inverse Association between Educational Status and Coronary CT Calcium Scores: Should We Reflect This in Our ASCVD Risk Assumptions?

**DOI:** 10.3390/ijerph20126065

**Published:** 2023-06-06

**Authors:** Christiane Dienhart, Bernhard Paulweber, Vanessa N. Frey, Bernhard Iglseder, Eugen Trinka, Patrick Langthaler, Elmar Aigner, Marcel Granitz, Bernhard Wernly

**Affiliations:** 1Department of Internal Medicine I, Paracelsus Medical University, 5020 Salzburg, Austria; 2Obesity Research Unit, Paracelsus Medical University, 5020 Salzburg, Austria; 3Department of Neurology, Christian Doppler Klinik, Paracelsus Medical University, 5020 Salzburg, Austria; 4Department of Geriatric Medicine, Christian Doppler University Hospital, Paracelsus Medical University, 5020 Salzburg, Austria; 5Department of Public Health, Health Services Research & Health Technology Assessment, UMIT—University for Health Sciences, Medical Informatics & Technology, 6060 Hall in Tirol, Austria; 6Centre for Cognitive Neuroscience, Neuroscience Institute, Christian Doppler University Hospital, Paracelsus Medical University, 5020 Salzburg, Austria; 7CT und MRT Institut, DBS Gmbh, 5020 Salzburg, Austria; 8Institute for General and Preventive Medicine, Paracelsus Medical University, 5020 Salzburg, Austria

**Keywords:** cardiovascular risk scores, educational status, SCORE2, cardiovascular health, CT calcium score

## Abstract

Education is not a factor included in most cardiovascular risk models, including SCORE2. However, higher education has been associated with lower cardiovascular morbidity and mortality. Using CACS as a proxy for ASCVD, we studied the association between CACS and educational status. Subjects, aged 40–69, from the Paracelsus 10,000 cohort, who underwent calcium scoring as part of screening for subclinical ASCVD, were classified into low, medium, and high educational status using the Generalized International Standard Classification of Education. CACS was dichotomised as either 0 or >0 for logistic regression modelling. Our analysis showed that higher educational status was associated with higher odds for 0 CACS (aOR 0.42; 95%CI 0.26–0.70; *p* = 0.001). However, there was no statistically significant association between the levels of total, HDL or LDL cholesterol and educational status, nor any statistical differences in HbA1c. SCORE2 did not differ between the three educational categories (4 ± 2% vs. 4 ± 3% vs. 4 ± 2%; *p* = 0.29). While our observations confirmed the relationship between increased educational status and lower ASCVD risk, the effect of educational status was not mediated via its impact on classical risk factors in our cohort. Thus, perhaps educational status should be taken into account to more accurately reflect individual risk in cardiovascular risk models.

## 1. Introduction

As acute complications of atherosclerotic cardiovascular disease (ASCVD) remain a leading cause of morbidity and mortality worldwide, it is imperative to identify individuals at risk of cardiovascular disease as early as possible in order to initiate appropriate risk reduction measures [1,2]. Current cardiovascular risk models do not include educational status as a risk factor for cardiovascular disease although several large US cohorts, including NHANES and ARIC, have evaluated the relationship between education level and ASCVD and have demonstrated a link between increased education and a lower risk of ASCVD [3,4]. A number of regional European studies have also substantiated this link [5,6,7,8].

Given the importance of educational status on general morbidity and mortality and particularly cardiovascular health, we felt it was important to evaluate this as an additional risk factor in the context of traditional risk factors such as metabolic syndrome, or smoking, hypertension, lipid levels, age and sex, contained within the context of SCORE2. Furthermore, given the interplay between educational status, socioeconomic status and health literacy, we expected that educational status could provide additional insights into the development of ASCVD.

Education is a complex determinant that influences a variety of health behaviors, including dietary choices, physical activity, tobacco and alcohol use, and adherence to medical treatments. These behaviors, in turn, significantly influence cardiovascular health and the risk of ASCVD. Incorporating educational status into the SCORE2 system could, therefore, improve its predictive ability for ASCVD risk. It could help us to understand patients’ health behaviors more comprehensively, allowing us to better predict who is at higher risk of ASCVD. Furthermore, understanding the educational disparities in ASCVD risk could provide opportunities for tailored interventions. For instance, people with lower educational status might require different types of interventions compared to those with higher educational status, such as more comprehensive lifestyle coaching or additional support in managing cardiovascular risk factors. This level of personalization could potentially lead to more effective prevention and management of ASCVD, ultimately improving the health outcomes of diverse populations.

Numerous studies have shown that CACS (computer tomography assessed coronary artery calcium score) values correlate with cardiovascular morbidity and mortality, as well as all-cause mortality [9,10,11]. CACS is performed non-invasively and can be used to diagnose calcium deposits in coronary vessels before these become symptomatic [12]. Furthermore, the radiation exposure of multi-slice computer tomography for coronary calcium diagnostics has been reduced to approximately 1 mS or less [13].

Both the American Heart Association (AHA) and the European Society of Cardiology (ESC) recommend the use of CACS in addition to a standard cardiovascular risk calculator to help assess cardiovascular risk status. The recently published DANCAVAS results further substantiate that cardiovascular screening, including CACS, blood pressure measurement and selected laboratory tests, plus treatment as appropriate, may lower the risk of death, heart attack and stroke particularly in patients under 70 years of age [14]. CACS measurement performed within the context of chest computer tomography (CT) to assess COVID-19 status has also increased the acceptance of CACS [11,15]. Nevertheless, there are currently no studies that have published substantial CACS data in an Austrian population. Furthermore, in Europe, there have been few studies specifically analysing the relationship between CACS, educational status and SCORE2. Thus, using CACS as a proxy for ASCVD, we have analysed the effects of educational status in our central European cohort, as we believe that educational status has the potential to add value to risk models and needs to be addressed not only by epidemiologists but also by clinicians in day-to-day practice.

## 2. Materials and Methods

Our analysis is based on cohorts derived from data gathered as a part of the Paracelsus 10,000 study [16]. The Paracelsus 10,000 study is a prospective local population-based study, in which a cohort of approximately 10,000 individuals aged 40 to 77 years from the Salzburg region were recruited randomly from a local population registry. The entire data set was collected between 2013 and 2020, and analysed retrospectively.

At baseline, all study participants were subjected to a screening program that included a detailed personal and family history, a physical examination and measurement of various anthropometric, clinical and laboratory parameters.

Approximately one out of five study participants underwent further intensive phenotypic characterization, including multi-slice CT scans to determine the coronary calcium score as calculated according to Agatston et al. [12].

Participants were stratified into the following three classes: low, medium and high educational status, based on the Generalised International Standard Classification of Education as described in the work of Schneider et al. (shown in Table 8), which provides an example of Austrian education levels coded according to GISCED and Edustrat [17]. GISCED 1 and 2, which were defined as subjects who completed only compulsory education, or less, were combined as ‘low’ educational status. Subjects who completed, at most, either vocational education and training at an upper secondary level, or, the Austrian equivalent of high school (Matura), i.e., categories 3 and 4, were classed as ‘medium’. Finally, subjects who completed at least post-high school level education, including vocational college or a bachelor’s degree (categories 5 and 6 according to Schneider et al.), were labelled as ‘high’ educational status.

Statistics were performed using Stata. Analyses were stratified by educational status (low, medium and high) to evaluate differences in baseline characteristics. Data are presented as mean ± standard deviation (SD) for continuous variables, and as number (*N*) and percent (%) for categorical variables. In order to analyse the effects of the educational status on CACS, we categorised calcium scores dichotomously as either 0 or >0 with a CACS = 0 as the dependent variable in the logistic regression models. Univariate as well as multivariable logistic regression were used to determine the relationship of CACS with educational status. We fitted the following three models: model-1 adjusted the primary exposure for age and sex, model-2 for age, sex and the concomitant diagnosis of a metabolic syndrome, and model-3 for age, sex and SCORE2, the ESC’s cardiovascular risk assessment tool [18] The models were height adjusted to take account of this possible confounding factor. We obtained adjusted odds ratios (aORs) and the respective 95% confidence intervals (CI). A *p*-value of <0.05 was considered significant. All tests were performed as two-sided.

## 3. Results

### 3.1. Population Demographics

Our study population (*N* = 1655) was nearly equally distributed between men (51.7%) and women (48.3%). The average age was 55 years. Our cohort is well educated, with the vast majority (94%) having at least a medium level of education. In the lowest level of education, women represented 61%, while at the highest level of education, men held the majority (55%). A further descriptive demographic overview is included in Table 1.

Our analysis showed no statistically significant association between the levels of total, HDL or LDL cholesterol and educational status, nor any statistical differences in HbA1c between the groups with low, medium or high educational status. SCORE2 also did not differ between the three educational categories (4 ± 2% vs. 4 ± 3% vs. 4 ± 2%; *p* = 0.29). Furthermore, the groups were not statistically different in terms of self-reported rates of diabetes, coronary artery disease, chronic heart failure, COPD and chronic kidney disease. Only self-reported levels of arterial hypertension showed a significant difference between the groups, with statistically lower levels of reported hypertension associated with higher education. Our data also show that the mean height increased in participants as educational status increased (168 ± 10 cm vs. 172 ± 9 cm vs. 173 ± 9 cm; *p* <0.001). The results from our study show an inverse relationship between education level and BMI, weight and abdominal circumference. An association between metabolic syndrome and educational status was borderline (*p* = 0.050).

### 3.2. Regression Analysis Results

A positive CACS is highly dependent on age and particularly associated with male sex (4–5 times higher odds than female sex) in our cohort. Therefore, we adjusted all models for age and sex. Nevertheless, after adjusting for age and sex, a higher level of education was still associated with higher odds for a negative CACS score (aOR 0.42; 95%CI 0.26–0.70 *p* = 0.001; Table 2).

In our multivariable regression models, using low educational status as a reference, subjects with medium (aOR 0.58 95%CI 0.0.36–0.91; *p* = 0.02) and higher (aOR 0.42 95%CI 0.26–0.70; *p* = 0.001) educational status evidenced lower odds for a positive CACS compared to subjects with lower education. Even after the adjustment for age, sex, height and metabolic syndrome, medium education status (aOR 0.66 95%CI 0.41–1.06; *p* = 0.08) reflected the tendency of lower odds for a positive CAC as education status increased, while a high education status was associated with a statistically significant reduction in the odds ratio (aOR 0.49 95%CI 0.30–0.82; *p* = 0.007). The relationship was maintained even after the adjustment for SCORE2 levels (Table 2).

## 4. Discussion

Using CACS as a proxy for ASCVD and GISCED guidelines to measure educational status, our analysis shows an inverse relationship between education status and ASCVD, in line with previous ASCVD studies in the US and Europe. However, our analysis also indicates that education may be an independent risk factor beyond the classical risk factors included in current AHA and ESC models. Of note, SCORE2 as well as metabolic risk factors did not adequately reflect this association between educational status and ASCVD. Hence, our data strongly suggest that the association between educational status and ASCVD risk is not mediated by the impact of education on classical ASCVD risk factors.

According to the ESC SCORE2 model supplementary data, Austria is considered to have an intermediate country risk for ASCVD with a cardiovascular disease mortality rate of 130.9 per 100,000 based on 2016 data [18]. According to Timmis et al. [19], the incidence of ischemic cardiovascular disease is 215/100,000, with a prevalence of 1788/100,000. In Austria, few analyses of the educational status effects on cardiovascular health have been published. In a superficial analysis published in 2011, Stein et al. [20] showed that the educational gradient in Austria is inverse to the east–west gradient in cardiovascular risk. In the same analysis based on data from 2009, the Salzburg region was shown to have a similar rate of cardiovascular deaths as the average in Austria. The rate of a positive CACS (>0) in our study was 25% for females and 38% for males. While we do not have follow-up data to date, not all of these subjects will develop cardiovascular morbidity and mortality. Nevertheless, with just over a third of all deaths in Austria due to cardiovascular disease [21], our figures reflect the scope of the problem.

OECD data suggest that Austria has an intermediate level of health inequality, although inequalities in the perceived unmet needs and unmet needs due to cost are both low [22]. Austria has a comparatively low level of income inequality based on the GINI index relative to the US. Austria is ranked #9 based on least income inequality, while the US is ranked #35 among OECD countries based on 2019 data [23].

It is hypothesized that the inverse relationship between health and education is associated with health and income inequalities related to the high education and insurance costs in the US, as well as the lack of access to health care caused by socioeconomic status differences [4]. Nevertheless, European studies have linked higher educational levels with improved health [6]. We consider educational status to be a valid surrogate parameter for socioeconomic status. Our finding that socioeconomically worse-off subjects also have worse health is consistent with prior literature [24].

A possible explanation for the maintained association between lower education status and increased ASCVD risk is poor health literacy. The American Heart Association published a scientific statement on the relevance of health literacy in primary and secondary prevention of cardiovascular disease in 2018 [25]. According to this statement, health literacy has an impact on both the prevention and management of ASCVD and it is recommended that clinicians both ‘anticipate and address’ problems caused by a lack of health literacy. Furthermore, in their statement, the AHA underlines the strong relationship between education, socioeconomic status and health literacy. According to the Health Literacy Survey in Europe, in 2011, over 55% of Austrians had inadequate or problematic health literacy scores [26]. The poor health literacy results generated government initiatives to improve access to health information, improve health related communication and promote health initiatives more effectively. In Austria, the initiative achieved a positive result. Although not directly comparable due to adjustments in questionnaires, in the latest survey, completed in 2020, just over 15% of Austrians had inadequate or problematic health literacy [27]. However, in the Austrian government report based on the recent survey, it was found that health literacy was particularly low in persons with low educational status and low socioeconomic status.

Another explanation for the correlation between education and ASCVD might be in the relationship between height and ASCVD risk and a further relationship between height and education. Our study results indicate a clear association between height and education level (*p* < 0.001). The relationship between shorter height and an increased risk of ASCVD is well-established and explained both by associations between shorter height and an adverse lipid profile, especially LDL (and triglycerides) as well as genetic analysis that indicates that some shared biologic paths may determine height and the development of ASCVD [28,29,30]. The relationship between height and educational status is also well-described in the literature [29,30,31,32]. For example, the worldwide Emerging Risk Factors Collaboration Study showed that in a cohort of over one million subjects, higher education was associated with an approximate 5 cm increase in height [30]. The reasons for the association between height and education are still unclear and probably multifactorial, ranging from genetic links to assortative mating, as well as interactions between genetic and environmental factors, including early life experiences and socioeconomic differences [33]. However, based on our results, the association between higher risk for ASCVD and lower education remained after extensive multivariable adjustment, including classical cardiovascular risk factors, as well as height. Therefore, we think that the link between educational status and CVD transcends both “traditional” cardiovascular risk factors as well as simple anthropometric measures, such as height.

While the link between ASCVD and educational status has been proven in a number of separate countries in Europe, we believe our results indicate that the country component of SCORE2 does not adequately reflect relative educational status. We believe that our analysis points to educational status as an important independent risk factor for ASCVD, which should be taken into account. Therefore, we hypothesize that the implementation of educational status as a proxy for socioeconomic status could improve the prediction of cardiovascular risk. In addition, tailored cardiovascular screening and risk reduction strategies could improve patient outcomes, particularly in the socioeconomically disadvantaged populations.

While metabolic syndrome showed a clear positive correlation with a CACS > 0, the association between educational status and ASCVD remained statistically significant after multivariable adjustment for the concomitant diagnosis of metabolic syndrome. Of note, SCORE2, which takes into account variables including age, sex, cholesterol and blood pressure, was not significantly different between groups with different education statuses. This may indicate that educational status may be a further risk factor for ASCVD, which is not adequately reflected in common risk models. Therefore, the relationship between ASCVD and educational status might go beyond the increase in cardiometabolic risk factors among socioeconomically weaker subjects.

One might argue that perhaps CACS in our population was not a good proxy for ASCVD, but we believe that the arguments for a positive CACS as a sign of ASCVD are clear. Furthermore, we appreciate that as a cross-sectional study, we have an association but not necessarily causality. It may be the case that educational status is purely a marker of increased risk, but not an actual factor. We look forward to analysing follow-up data to better correlate our current CACS results with actual cardiovascular morbidity and mortality rates. While we believe our cohort is large enough to avoid confounding, it will be important to better understand the drivers behind the relationship between educational status and ASCVD. However, further elucidation of the components driving the relationship between educational status and ASCVD is beyond the scope of this study. Furthermore, unfortunately, without actual health literacy surveys on our patients, we can only make assumptions on the impact of health literacy as a potential explanation for our observations. Nevertheless, we think that our study could serve in generating hypotheses for future studies in this field. For example, previous studies have shown a link between positive dietary choices such as an increased intake of fiber and reduced intake of refined sugars and starch with higher educational status [34,35]. Given the higher BMI and abdominal circumference associated with lower educational status demonstrated in our study, we believe this association merits further analysis. Particularly, longitudinal studies including accurately characterized nutritional data, among other known risk factors, are needed to further illuminate the complex relationship between educational status and ASCVD.

## 5. Conclusions

Our data show that an increasing education level, particularly a high level of education, is associated with a negative CACS, and thus a lower ASCVD risk in our Austrian cohort. While higher educational status is associated with a lower BMI, lower abdominal girth and decreased central obesity, our data do not show an association between educational status and classical ASCVD risk factors, including lipid levels, blood glucose measurements, or triglycerides. The association is maintained even when correcting for metabolic syndrome. Furthermore, as SCORE2 did not vary between groups, adding educational status to risk models might improve SCORE2 predictions.

Moreover, by acknowledging differences in educational status and related health literacy, clinicians may tailor their consultations more effectively to individual patients when addressing ASCVD risk. In addition to raising general awareness of the impact of educational status on ASCVD, this issue needs to be highlighted especially in the face of further digitization and the increasing need for electronic literacy. However, it is important to note that further research is needed to clarify how exactly educational status should be incorporated into ASCVD risk prediction models, and to what extent it improves their predictive accuracy. Beyond health literacy, other possible factors, such as dietary differences, which may be mediators of the association between educational status and ASCVD, need to be identified and examined in further studies.

## Figures and Tables

**Table 1 ijerph-20-06065-t001:** Descriptive overview of 1665 subjects with CACS in Paracelsus 10.000.

	GISCED = Low	GISCED = Medium	GISCED = High	*p*-Value
Total number of subjects *N* = 1665	*N* = 96	*N* = 1194	*N* = 375	
Age (years)	55 (4)	55 (3)	55 (4)	0.66
Age 40–49 years	5% (5)	2% (26)	3% (12)	
Age 50–59 years	79% (76)	86% (1028)	84% (316)	
Age 60–69 years	16% (15)	12% (140)	13% (47)	
Hba1c %	5.5 (0.3)	5.5 (0.4)	5.5 (0.4)	0.31
Total cholesterol (mg/dl)	218 (42)	214 (38)	217 (39)	0.29
Triglycerides (mg/dl)	128 (81)	121 (81)	115 (62)	0.23
HDL cholesterol (mg/dl)	62 (16)	64 (18)	64 (18)	0.54
LDL cholesterol (mg/dl)	148 (41)	143 (37)	146 (36)	0.28
Height (cm)	168 (10)	172 (9)	173 (9)	<0.001
Weight (kg)	78 (16)	79 (16)	76 (14)	0.013
BMI (kg/m^2^)	28 (6)	27 (5)	26 (4)	<0.001
Abdom. circumference (cm)	94 (13)	94 (13)	91 (12)	<0.001
Self reported diagnosis:				
Dyslipidemia	10% (10)	11% (127)	10% (38)	0.96
Art. hypertension	28% (27)	21% (243)	15% (57)	0.01
Coronary artery disease	2% (2)	2% (21)	1% (5)	0.82
Chronic heart failure	0% (0)	0% (5)	0% (0)	0.37
Peripheral artery disease	1% (1)	0% (4)	0% (0)	0.23
COPD	2% (2)	2% (22)	2% (7)	0.99
Chronic kidney disease	1% (1)	0% (3)	1% (2)	0.38
BMI categories				
BMI < 18.5	1% (1)	1% (9)	1% (2)	
BMI 19.5 to 24.9	40% (38)	37% (443)	46% (174)	
BMI 25 to 29.9	27% (26)	40% (479)	40% (150)	
BMI 30 to 34.9	23% (22)	16% (197)	10% (38)	
BMI 35 to 39.9	7% (7)	4% (52)	3% (10)	
BMI ≥ 40	2% (2)	1% (14)	0% (1)	
CACS by educational status				
CAC 0	55% (53)	61% (728)	66% (246)	
CAC 1–300	36% (35)	33% (397)	30% (113)	
CAC > 300	8% (8)	6% (69)	4% (16)	
Metabolic syndrome *	20% (18)	15% (182)	11% (42)	0.050
Educational status by gender				
Men	39% (37)	52% (616)	55% (207)	
Women	61% (59)	48% (578)	45% (168)	

* Metabolic syndrome according to the International Diabetes Federation Criteria.

**Table 2 ijerph-20-06065-t002:** Logistic regression models using low educational status as reference; height adjustment.

Educational Status	Model-1 ^1^	Model-2 ^2^	Model-3 ^3^
Low	ref	ref	ref
Medium	aOR 0.58; 95%CI 0.36–0.91; *p* = 0.02	aOR 0.66; 95%CI 0.0.41–1.06; *p* = 0.08	aOR 0.65; 95%CI 0.41–1.05; *p* = 0.08
High	aOR 0.42; 95%CI 0.26–0.70; *p* = 0.001	aOR 0.49; 95%CI 0.30–0.82; *p* = 0.007	aOR 0.50; 95%CI 0.30–0,84; *p* = 0.009

^1^ Model-1: adjustment for age and sex. ^2^ Model-1: adjustment for age, sex, height and metabolic syndrome. ^3^ Model-1: adjustment for age, sex, height and SCORE2.

## Data Availability

Data are available upon reasonable request and in compliance with European Data protection regulations. Please contact Vanessa Frey for details at v.frey@salk.at.

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
