# Peer review of "Inverse Association between Educational Status and Coronary CT Calcium Scores: Should We Reflect This in Our ASCVD Risk Assumptions?"

_ijerph, 2023, doi:10.3390/ijerph20126065_

Round 1
Reviewer 1 Report
The authors found an association between coronary calcium and educational level, but not with traditional risk factors. However, there was a relationship with obesity and education levels. SCORE2 only takes into account some risk factors, but not all. Therefore, education could be a risk marker (eating worse, more obesity, etc.), rather than a risk factor. And this could not be differentiated by the study. On the other hand, the authors associated education levels with coronary calcium in a cross-sectional study. In order to find out if educational level is really a risk factor, a longitudinal study would have to be carried out, as more fators could have been controlled.
Author Response
Thank you for taking the time to review our manuscript. We appreciate the opportunity to enhance our paper, as well as our further research based on your very useful, although critical comments. We fully understand your reservations and have highlighted these in the discussion when addressing limitations. We absolutely agree that SCORE2 does not take into account enough risk factors and that perhaps taking more risk factors into account would improve risk prediction. Educational status has been associated with ASCVD risk in a number of published studies. We believe that educational status may be a marker for socioeconomic status as well as reflect health literacy. It is, of course, a fair point that it may be a risk marker rather than a risk factor although we would argue that based on the English definition from Merriam Webster or Oxford English dictionary, we may use the term 'factor' for educational status as it has, according to our analysis, an influence on CACS. However, we have integrated this important consideration that educational status may be purely a risk marker and not a risk factor into our discussion. For the most part, we have used conditional phrasing and the term ‘association’ to describe the relationship between educational status and ASCVD as we have tried not to be misleading. We would have loved to be able to have analysed longitudinal data, but we do not have this yet.
We have attached a file addressing your feedback (see reviewer 1) which might be more easily readable. Furthermore, it gives our response to the other reviewers to provide additional context.
Reviewer 2 Report
The passage highlights the fact that education is not considered in most cardiovascular risk models, such as SCORE2. However, studies have shown that higher education is associated with lower cardiovascular morbidity and mortality. The authors used Coronary Artery Calcium Scoring (CACS) as a proxy for Atherosclerotic Cardiovascular Disease (ASCVD) to study the association between CACS and educational status. The study included subjects aged 40-69 from the Paracelsus 10,000 cohort, who underwent calcium scoring as part of a screening process for subclinical ASCVD. In conclusion, this study provides valuable insights into the potential association between educational status and cardiovascular risk. The findings emphasize the need for further investigation into the role of education in cardiovascular risk models, as well as the possible underlying mechanisms that may contribute to this association. Incorporating educational status into cardiovascular risk models could potentially lead to more accurate assessments of individual risk, ultimately improving prevention and intervention strategies for cardiovascular diseases.
There are some places need to be addressed:
1. Please clarify in line 87 what e Generalised International Standard Classification of Education will be cited as “GISCED”.
2. The authors mention poor health literacy as a possible explanation for the maintained association between lower education status and increased ASCVD risk. It would be interesting to see if the authors could explore this relationship in more depth or provide suggestions for future research on how improving health literacy could potentially impact ASCVD risk.
3. The study found a significant association between height and education level. It would be helpful for the authors to clarify whether this association was controlled for in their analyses or if further investigation is needed to determine the impact of height on the relationship between educational status and ASCVD risk.
4. The authors mention the link between positive dietary choices and higher educational status, suggesting that this association should be analyzed in further studies. It would be helpful for the authors to provide more information on how they plan to investigate this relationship and the potential implications for ASCVD risk prediction and management.
Overall, the study is well-executed, and the findings have significant implications for understanding the role of educational status in ASCVD risk. Addressing the above points will help to strengthen the manuscript and provide a clearer direction for future research in this area.
Author Response
Thank you very much for taking the time to review our manuscript. Thank you very much for your kind feedback and very valuable comments. We greatly appreciate your constructive feedback and the opportunity to enhance our manuscript based on your comments.
Below is a detailed response specifically to your comments. The answers are also addressed in the file attached under 'reviewer 2' which may be more easily readable. However, the document also contains our responses to all the reviewer comment.
- Please clarify in line 87 what Generalised International Standard Classification of Education will be cited as “GISCED”.
We have added the following to clarify (see line100):
as described in Schneider at al. table 8, which provides an example of Austrian education levels coded according to GISCED and Edustrat.[17] GISCED 1 and 2, which were defined as subjects who completed only compulsory education, or less, were combined as ‘low’. Subjects who completed, at most, either vocational education and training at an upper secondary level, or, the Austrian equivalent of high school (Matura), i.e. categories 3 and 4, were classed as ‘medium’. Finally, subjects who completed at least a post high school level education, including vocational college or bachelor (classes 5 and 6), were labelled as ‘high’ educational status.
- The authors mention poor health literacy as a possible explanation for the maintained association between lower education status and increased ASCVD risk. It would be interesting to see if the authors could explore this relationship in more depth or provide suggestions for future research on how improving health literacy could potentially impact ASCVD risk.
We have added the following:
The American Heart Association published a scientific statement on the relevance of health literacy in primary and secondary prevention of cardiovascular disease in 2018. [25] According to this statement, health literacy has an impact on both prevention and management of ASCVD and it is recommended that clinicians both ‘anticipate and address’ problems caused by lack of health literacy. Furthermore, in their statement, the AHA underline the strong relationship between education, socioeconomic status and health literacy.
- The study found a significant association between height and education level. It would be helpful for the authors to clarify whether this association was controlled for in their analyses or if further investigation is needed to determine the impact of height on the relationship between educational status and ASCVD risk.
We adjusted for height in our original analysis (see footnote for Models 2 and 3 in Table 2) and found the relationship between educational status and ASCVD risk continued to be significant even adjusted for height. We mention this in our paper (see lines 157 and 236)
The authors mention the link between positive dietary choices and higher educational status, suggesting that this association should be analyzed in further studies. It would be helpful for the authors to provide more information on how they plan to investigate this relationship and the potential implications for ASCVD risk prediction and management.
Thank you very much for your constructive comment and for highlighting this area of interest in our study. We agree with you that the association between positive dietary choices and higher educational status indeed warrants further research, particularly regarding its potential implications for ASCVD risk prediction and management. Your suggestion to elaborate on our plans for future investigation is indeed pertinent. However, at this point in time, our study was purely observational and we did not collect additional data that would enable us to investigate this relationship in more depth. Given the importance of this topic, we are currently in the process of drafting new research proposals that would delve further into this complex relationship. We appreciate your insightful feedback, which certainly adds value to the broader discussion about diet, education, and health outcomes. We hope that our future research will address this question and contribute valuable insights into ASCVD risk prediction and management.
Reviewer 3 Report
1. Interesting analysis. Introduction can mention the reason for studying education as a factor. Is it because of the availability of CACS?
2. What is the CVD mortality in Austria? Variation by educational status? By socio-economic factors?
3. Has any other studies demonstrated such independent effect?
4. Is the study robust to enough to address confounding.
5. What are the risk factors for CACS?
6. Can this replace more complex CVD risk scores? Can people be managed based on CACS? If they also need full assessment, then predicting CACS by education may not add much value?
7. Propose a method to study this further.
Author Response
Thank you very much for taking the time to review our manuscript. Thank you very much for your feedback and very valuable comments. We greatly appreciate your constructive feedback and the opportunity to enhance our manuscript based on your comments.
Below is our response to your feedback. We have also provided our responses in the file attached which may be more easily readable. Our specific response to your feedback is in the section 'reviewer 3'. The document however also provides our responses to the other reviewers for further context.
- Interesting analysis. Introduction can mention the reason for studying education as a factor. Is it because of the availability of CACS?
We have added the following to our introduction:
Given the importance of educational status on general morbidity and mortality and particularly cardiovascular health, we felt it was important to evaluate this as an additional risk factor in the context of traditional risk factors such as metabolic syndrome, or those contained within the context of SCORE2 such as smoking, hypertension, lipid levels, age and sex. Furthermore, given the interplay between educational status, socioeconomic status and health literacy, we expected that educational status could provide additional insights into the development of ASCVD.
Education is a complex determinant that influences a variety of health behaviors, including dietary choices, physical activity, tobacco and alcohol use, and adherence to medical treatments. These behaviors, in turn, significantly influence cardiovascular health and the risk of ASCVD. Incorporating educational status into the SCORE2 system could therefore improve its predictive ability for ASCVD risk. It could help us understand patients' health behaviors more comprehensively, allowing us to better predict who is at higher risk of ASCVD. Furthermore, understanding the educational disparities in ASCVD risk could provide opportunities for tailored interventions. For instance, people with lower educational status might require different types of interventions compared to those with higher educational status, such as more comprehensive lifestyle coaching or additional support in managing cardiovascular risk factors. This level of personalization can potentially lead to more effective prevention and management of ASCVD, ultimately improving the health outcomes of diverse populations. However, it is important to note that further research is needed to clarify how exactly educational status should be incorporated into ASCVD risk prediction models, and to what extent it improves their predictive accuracy. Thank you for your insightful question. We look forward to further investigating this topic in our future research when and if additional data is available.
What is the CVD mortality in Austria? Variation by educational status? By socio-economic factors?
Unfortunately, there is very little reliable data, which is why we feel our study adds value. However, we have tried to answer your question and add more context to the discussion as follows:
According to the ESC SCORE2 model supplementary data, Austria is considered to have an intermediate country risk for ASCVD with a cardiovascular disease mortality of 130.9 per 100,000 based on 2016 data [18]. According to Timmis [19] et al, the incidence of ischemic cardiovascular disease is 215/100,000, with a prevalence of 1,788/100,000. In Austria, there have been very few analyses of educational status effects on cardiovascular health. In a superficial analysis published in 2011, Stein et al.[ref] showed that the educational gradient in Austria is inverse to the east-west gradient in cardiovascular risk. In the same analysis based on data from 2009, the Salzburg region was shown to have a similar rate of cardiovascular deaths as the average in Austria.
3. Have any other studies demonstrated such independent effect?
See answer above with respect to Austria. The effect of education on the risk of ASCVD has been shown in multiple US and European studies (mentioned in Introduction: …including NHANES and ARIC, have evaluated the relationship between education level and ASCVD and have demonstrated a link between increased education and a lower risk of ASCVD.[3,4] A number of regional European studies have also substantiated this link. [5–8]). However, in Austria we have a comparatively well-educated population and lower health care inequality) so the ‘educational status’ effect is something beyond these, and, according to our data, additive to traditional risk factors.
Is the study robust to enough to address confounding.
We believe, the number of subjects is statistically adequate, however, confounding can never truly be ruled out. There are also, of course, issues with a cross sectional study. We have tried to address this more clearly in the paragraph on limitations. We hope to address these issues in future studies when we are able to analyse the data on factors with may interplay with educational status, such as diet, as well as when we have longitudinal data.
What are the risk factors for CACS?
The traditional risk factors for ASCVD, such as age, male sex, hypertension, metabolic syndrome, smoking, and elevated LDL are also risk factors for a positive CACS. These factors are taken into account in traditional risk models including SCORE2, which is recommended for use in Europe. CACS are predictive of ASCVD as indicated for example by the recommendation of the ESC to use a positive CACS to upgrade ASCVD risk in patients. We feel that educational status would be a good proxy for socioeconomic status as well as health literacy, which the SCORE2 risk calculator does not take into account.
Can this replace more complex CVD risk scores? Can people be managed based on CACS? If they also need full assessment, then predicting CACS by education may not add much value?
This is a very thoughtful and interesting question, which is however, very difficult to answer and, we believe, beyond the scope of our study. The ESC suggests that CACS can be used to upgrade (and downgrade) risk particularly for patients who are categorised as intermediate risk using the risk model, SCORE2 (2b evidence). The AHA also suggests similar use of CACS. While Ties at al suggested in the European Heart Journal (2023) that CACS should be applied more generously to avoid missing patients with high CACS, a 2022 paper in JAMA by Bell at al. argued that there is only a modest incremental benefit of a CACS which needs to be weighed against the cost (including incidental findings) and possible radiation risk. Furthermore, the gain may be smaller for risk scores that take a larger number of factors into account (for example QRISK3). It was not our intention to suggest that a CACS should be used to replace a more complex CVD risk score, however, we believe that the evidence is strong enough to allow the use of CACS as a proxy for ASCVD in order to understand risk factors within population studies. While we have shown a relationship between CACS and traditional risk factors, we would argue that education should be included in risk scores in some way as it reflects an added risk that is not taken into account in most models. Furthermore, clinicians should consider education when consulting with a patient as education could affect health literacy.
Propose a method to study this further.
We look forward to analysing follow-up data from our study in coming years to study how closely our CACS actually predict cardiovascular morbidity and mortality. Knowing how to apply CACS in both a country context as well as by age and gender is important to meeting the ESC recommendations. Furthermore, we hope to study the educational status effect in more detail in order to understand why educational status makes such a difference. There is good evidence from various studies that, for example, educational status has an effect on the composition of diet. Anecdotally, I would tend to agree that, for example, builders in my diabetes and lipid clinics tend to drink much more sweetened drinks and eat much more processed meat and less vegetables but, we would need to analyse very complex data from our population to hopefully prove such an association. We have added a comment to our discussion for the need for longitudinal studies which stratify patients by educational status and which have detailed data both for nutrition and other risk factors.
Round 2
Reviewer 1 Report
Authors are properly answered the suggestions performed.